# New Perspective on Planar Inductive Sensors: Radio-Frequency Refractometry for Highly Sensitive Quantification of Magnetic Nanoparticles

**DOI:** 10.3390/s23052372

**Published:** 2023-02-21

**Authors:** José Luis Marqués-Fernández, María Salvador, José Carlos Martínez-García, Pablo Fernández-Miaja, Alfredo García-Arribas, Montserrat Rivas

**Affiliations:** 1Department of Physics & IUTA, University of Oviedo, Campus de Viesques, 33203 Gijón, Spain; 2Department of Electrical Engineering, University of Oviedo, Campus de Viesques, 33203 Gijón, Spain; 3Department of Electricity and Electronics, University of the Basque Country, 48940 Leioa, Spain

**Keywords:** self-resonant frequency, inductive sensor, coil, nanoparticles, magnetic nanoparticles, magnetic lateral flow immunoassays, impedance, refraction index, magnetic permeability, electric permittivity

## Abstract

We demonstrate how resonant planar coils may be used as sensors to detect and quantify magnetic nanoparticles reliably. A coil’s resonant frequency depends on the adjacent materials’ magnetic permeability and electric permittivity. A small number of nanoparticles dispersed on a supporting matrix on top of a planar coil circuit may thus be quantified. Such nanoparticle detection has application detection to create new devices to assess biomedicine, food quality assurance, and environmental control challenges. We developed a mathematical model for the inductive sensor response at radio frequencies to obtain the nanoparticles’ mass from the self-resonance frequency of the coil. In the model, the calibration parameters only depend on the refraction index of the material around the coil, not on the separate magnetic permeability and electric permittivity. The model compares favourably with three-dimensional electromagnetic simulations and independent experimental measurements. The sensor can be scaled and automated in portable devices to measure small quantities of nanoparticles at a low cost. The resonant sensor combined with the mathematical model is a significant improvement over simple inductive sensors, which operate at smaller frequencies and do not have the required sensitivity, and oscillator-based inductive sensors, which focus on just magnetic permeability.

## 1. Introduction

New sensors are needed to detect and quantify nanoparticles (NPs) for increasing new applications. One example is the detection of biomolecules (toxins, disease biomarkers, or others), which are selectively attached to the particles through an immunological reaction [1]. There are several approaches for it, such as immobilising the particle-labelled molecule onto the sensor surface, using microfluidics to take the sample to the sensor or to pass it over its sensitive part, or paper-based microfluidics [2,3]. The latter is well known for being the basis of the COVID-19 rapid diagnostic tests [4]. Such tests have two main limitations that hinder their wider application; one is the lack of sensitivity (many false-negative results), and the other is their unreliable quantification. To improve it, some authors rely on magnetic nanoparticles (MNPs) as labels [5,6,7,8], which can be detected by devices sensitive to the fringes of their magnetic field, as in [9]. New needs in life sciences, such as healthcare, food safety, and environmental control, require further improvement in the sensitivity and quantifying capabilities of MNP-detecting sensors due to the extremely small number of particles immobilised at the test line and their tiny size.

Previous works [10,11,12] highlighted the ability to detect and quantify small numbers of MNPs by monitoring the impedance changes in a planar coil at a fixed frequency in the order of tens of MHz. The sensor is sensitive to magnetic permeability variations around the planar coil [13] as a consequence of Faraday’s electromagnetic induction law and can be observed as changes in its impedance. The larger the frequency, the larger the impedance change, as long as we are below the coil’s resonance frequency. In general, reliably obtaining small impedance changes in the MHz range requires expensive equipment (such as impedance or vector-network analysers,) so lower-cost and portable solutions would be more accessible and better combined with the inexpensive LFIAs.

Some reported miniaturised devices focus on cost reduction using oscillator-based inductive detectors, and their reported sensitivity is 3 Hz/μg Fe${}_{3}$O${}_{4}$ [9,14]. Such sensitivity is not enough for the requirements of many new potential applications, such as toxin detection in foods or early diagnosis.

Based on the central idea of [14], in this work, we improved MNP detection sensitivity while keeping costs down using the sensor coils’ self-resonant frequency (SRF). The main roadblock of SRF-based detectors is the complexity of the mathematical relation between the SRF and MNPs’ mass. We solved this bottleneck as described in Section 2. A usable, easy-to-implement, and precise mathematical model to correlate the coils’ SRF and the nanoparticles’ mass (volume or number, depending on the variable used for the calibration) is indispensable to make this measurement technique viable. We developed such a targeted model and tested it against simulations and experimental results to verify its validity and explored its limitations. Even when the central purpose was to develop and test the mathematical model, our analysis revealed a significant sensitivity and signal-to-noise ratio increases.

## 2. Mathematical Fundamentals

The method we developed to quantify nanoparticles (NPs) involves a planar coil and monitoring its SRF for different NP masses (whether magnetic or not) placed on top of it. Generally, the sensors that detect metal particles use single- or multiple-turn planar coils [15]. Due to its electromagnetic-field uniformity, we used a two-turn coil for this work. The planar nature of the coil allows the placement of flat samples (as the LFIAs) on top of the sensor for its measurement. Figure 1 illustrates a bi-dimensional projection of the coil shape.

The main goal of this section is to deduce a mathematical expression that relates the planar coil’s SRF and the NP mass on top of it. Using some hypotheses, we modelled this relationship in a simple and easy-to-use manner. The first simplifying hypothesis is to consider only the first SRF mode of our sensing coil, which allows us to model the coil’s impedance as an equivalent circuit. Using equivalent circuits to define the behaviour of transmission lines is a widespread and accepted procedure [16,17,18,19]. Simple inductors, such as those in Figure 1 are generally modelled using an inductor–capacitive network, as in Figure 2.

The model has been used before to approximate the core material properties of the coil [20], including complex magnetic permeability and electric permittivity as follows: way.
(1)$$Z_{L}=\omega Li\left(\mu_{\mathrm{eff}}^{{}^{\prime }}-i\mu_{\mathrm{eff}}^{{}^{\prime \prime }}\right)$$
(2)$$Z_{C}=\frac{1}{\omega Ci(\varepsilon_{\mathrm{eff}}^{{}^{\prime }}-i\varepsilon_{\mathrm{eff}}^{{}^{\prime \prime }})}$$
(3)$$Z_{\mathrm{EQ}}={\left(\frac{1}{Z_{L}+Z_{R}}+\frac{1}{Z_{C}}\right)}^{-1}$$
where $Z_{L}$ stands for the inductive component, $Z_{C}$ is the capacitive one, and $Z_{R}=R$ is the resistance of the equivalent circuit; ω is the operating angular frequency, *L* is the coil self-inductance, and *C* its capacitance; $\left(\mu_{\mathrm{eff}}^{{}^{\prime }}-i\mu_{\mathrm{eff}}^{{}^{\prime \prime }}\right)$ and $\left(\varepsilon_{\mathrm{eff}}^{{}^{\prime }}-i\varepsilon_{\mathrm{eff}}^{{}^{\prime \prime }}\right)$ are, respectively, the effective complex initial magnetic permeability and effective complex initial electric permittivity. Here, the term “effective” and the subindex “eff” refer to the effect of all the materials surrounding the coil. Equation (Equation 3) has some frequency-dependent parameters, such as the inductive and capacitive reactances, material properties (for example, the magnetic permeability is frequency-dependent, as established with the Neél or Brownian model, depending on the combination of material composition and size, magnetic-field intensity, and its frequency range), skin effects (which, being a consequence of Faraday’s induction, narrows the current path with increasing frequencies), and other non-linearities. Regardless, in the proximities of the SRF, NPs produce only minor variations in the frequency, so, in a first approximation, these variables can be taken as constants. The equivalent impedance model allows us to calculate the SFR as the first frequency value for which the impedance’s imaginary component vanishes. This approach has been previously used to characterise choke cores [21].
(4)$$\mathrm{SRF}=\frac{\sqrt{LC\;\varepsilon_{\mathrm{eff}}^{{}^{\prime }}\;{\mu_{\mathrm{eff}}^{{}^{\prime }}}^{3}-C^{2}\;{Z_{R}}^{2}\;{\varepsilon_{\mathrm{eff}}^{{}^{\prime }}}^{2}\;{\mu_{\mathrm{eff}}^{{}^{\prime }}}^{2}+LC\;\varepsilon_{\mathrm{eff}}^{{}^{\prime }}\;{\mu_{\mathrm{eff}}^{{}^{\prime \prime }}}^{2}\;\mu_{\mathrm{eff}}^{{}^{\prime }}}-C\;Z_{R}\;\varepsilon_{\mathrm{eff}}^{{}^{\prime }}\;\mu_{\mathrm{eff}}^{{}^{\prime \prime }}}{2\pi CL\;\varepsilon_{\mathrm{eff}}^{{}^{\prime }}\;{\mu_{\mathrm{eff}}^{{}^{\prime }}}^{2}+2\pi LC\;\varepsilon_{\mathrm{eff}}^{{}^{\prime }}\;{\mu_{\mathrm{eff}}^{{}^{\prime \prime }}}^{2}}$$

Given that, in general, the number of NPs to quantify is meagre, we expect that $\mu_{\mathrm{eff}}^{{}^{\prime \prime }}\approx 0$. In any case, the numerical simulations prove this assumption to be acceptable. For our coil geometry, $C^{2}Z_{R}^{2}$ is much smaller than LC so that the expression of the SRF can be simplified to
(5)$$\mathrm{SRF}=\frac{\sqrt{LC\;\varepsilon_{\mathrm{eff}}^{{}^{\prime }}\;{\mu_{\mathrm{eff}}^{{}^{\prime }}}^{3}}}{2\pi LC\;\varepsilon_{\mathrm{eff}}^{{}^{\prime }}\;{\mu_{\mathrm{eff}}^{{}^{\prime }}}^{2}}=\frac{1}{2\pi \sqrt{LC\mu_{\mathrm{eff}}^{{}^{\prime }}\varepsilon_{\mathrm{eff}}^{{}^{\prime }}}}=\frac{1}{2\pi \sqrt{LC}n_{\mathrm{eff}}^{{}^{\prime }}}$$
where $\sqrt{LC}$ is a constant (taking into account the described assumption) characteristic of the coil, and $n_{\mathrm{eff}}^{{}^{\prime }}$ is the real part of the effective refraction index $n_{\mathrm{eff}}^{{}^{\prime }}=\sqrt{\mu_{\mathrm{eff}}^{{}^{\prime }}\varepsilon_{\mathrm{eff}}^{{}^{\prime }}}$, which considers the contribution of all the materials around the coil.

Determining the effective refraction index of a composite medium is a complex problem explored in detail in the effective medium theory (EMT) [22]. In this context, the volume distribution of materials, their properties, and the electromagnetic field’s spatial distribution determines the effective values of electrical permittivity and magnetic permeability. Given the high complexity, we need some simplification to provide a compact expression. For this purpose, we selected Lichtenecker’s model [23,24], which provides enough accuracy to model the effective refraction index near our sensor with the required simplicity.

Instead of mathematically modelling the sample on top of our sensor realistically as a prism, we modelled it as a uniform mass distribution surrounding the coil track, as shown in Figure 3. In the figure, the NPs are represented as brown dots, and the matrix material as blue dots. On the left, we can see the NPs mingled with the matrix in the shape of a prism, whereas the same amounts of both materials are spread around the coil in the right figure, as used in mathematics.

In this way, using Lichtenecker’s logarithmic mixing formula, one can model the effective magnetic permeability and the effective electric permittivity with two formally identical expressions,
(6)$$\begin{array}{cc} & \ln\left(\mu_{\mathrm{eff}}\right)=f\ln\left(\mu_{s}\right)+\left(1-f\right)\ln\left(\mu_{\mathrm{env}}\right) \end{array}$$
(7)$$\begin{array}{cc} & \ln\left(\varepsilon_{\mathrm{eff}}\right)=f\ln\left(\varepsilon_{s}\right)+\left(1-f\right)\ln\left(\varepsilon_{\mathrm{env}}\right) \end{array}$$
which enables further simplification, as will be shown in the next section. Here, *f* is the volume fraction occupied by the sample concerning the volume where the magnitude of the electromagnetic field is larger than a given threshold; $\mu_{\mathrm{eff}}$ and $\varepsilon_{\mathrm{eff}}$ are the effective permeability and permittivity, respectively, determined from the values $\mu_{s}$ and $\varepsilon_{s}$ corresponding to the sample, and $\mu_{\mathrm{env}}$ and $\varepsilon_{\mathrm{env}}$ for the environment. In turn, $\mu_{s}$ and $\varepsilon_{s}$ are related to the mass of nanotags and their support.

Using the logarithmic Lichtenecker’s mixing formula again results in
(8)$$\begin{array}{cc} & \ln\left(\mu_{s}\right)=\left(\left(\frac{m_{\mathrm{NPs}}}{m_{\max}}\right)\ln\left(\mu_{\mathrm{NPs}}\right)\right)+\left(\left(1-\left(\frac{m_{\mathrm{NPs}}}{m_{\max}}\right)\right)\ln\left(\mu_{\mathrm{mtx}}\right)\right) \end{array}$$
(9)$$\begin{array}{cc} & \ln\left(\varepsilon_{s}\right)=\left(\left(\frac{m_{\mathrm{NPs}}}{m_{\max}}\right)\ln\left(\varepsilon_{\mathrm{NPs}}\right)\right)+\left(\left(1-\left(\frac{m_{\mathrm{NPs}}}{m_{\max}}\right)\right)\ln\left(\varepsilon_{\mathrm{mtx}}\right)\right) \end{array}$$

As the NPs and the support are incompressible, $m_{\mathrm{NPs}}/m_{\max}$ is the volumetric relation, expressed as the ratio of NP mass in the sample and a reference mass corresponding to the same volume filled by NPs, with no voids; $\mu_{\mathrm{NPs}}$, $\varepsilon_{\mathrm{NPs}}$, $\mu_{\mathrm{mtx}}$, and $\varepsilon_{\mathrm{mtx}}$ are the values of the magnetic permeability and electric permittivity of the NPs and the matrix, respectively.

Depending on the measurement procedure, there may be no matrix, but NPs are deposited on a sample holder or directly on the sensing element. The combined environment refraction index accounts for the sample support and matrix, $n_{\mathrm{mtx}}=1$. Using this simplification and combining Equations (6)–(9), we obtain the following effective refraction index:(10)$$n_{\mathrm{eff}}=n_{\mathrm{env}}{\left(\frac{n_{\mathrm{NPs}}}{n_{\mathrm{env}}}\right)}^{f\left(\frac{m_{\mathrm{NPs}}}{m_{\max}}\right)}$$

Substituting $n_{\mathrm{eff}}$ in Equation (Equation 5) leads to a final expression that correlates the SRF to the mass of NPs in the sample as follows:(11)$$\mathrm{SRF}=\frac{1}{2\pi n_{\mathrm{env}}^{\prime }\sqrt{LC}{\left(\frac{n_{\mathrm{NPs}}^{\prime }}{n_{\mathrm{env}}^{\prime }}\right)}^{\frac{f\;m_{\mathrm{NPs}}}{m_{\max}}}}$$

Using natural logarithms, expression (Equation 11) leads to
(12)$$\ln\left(\mathrm{SRF}\right)=m_{\mathrm{NPs}}\;A_{1}-A_{0}$$
where
(13)$$\begin{array}{cc} & A_{1}=-\;\frac{f}{m_{\max}}\ln\left(\frac{n_{\mathrm{NPs}}^{\prime }}{n_{\mathrm{env}}^{\prime }}\right) \end{array}$$
(14)$$\begin{array}{cc} & A_{0}=\ln\left(2\pi \sqrt{L\;C}\;n_{\mathrm{env}}^{\prime }\right) \end{array}$$

One needs to remark that $A_{1}$ is a function of the relative relation between the NPs ($n_{\mathrm{NPs}}^{\prime }$) and their environment ($n_{\mathrm{env}}^{\prime })$, while $A_{0}$ depends on the characteristics of the coil (LC) and the environment. In this way, we can calibrate any sensor by determining $A_{1}$ and $A_{0}$ through a linear fit of the law in (Equation 12) using several NP masses and the corresponding experimental SRFs. This calibration allows us to quantify unknown NP masses in the same device. As our sensor sensitivity is determined by the ratio between the effective refractive indexes of the sample and the environment, we defined this method as “refractometry sensing at radio-frequency”.

## 3. Simulation and Experimental Procedures

The mathematical model obtained from the targeted hypothesis needs verification. We sought to corroborate the working principle of radio-frequency refractometry, the validity of the hypothesis, and the real use of the model. The working principle and corroboration of the hypothesis are better tested using simulation in which we have absolute control over the defining measuring parameters, and it is possible to test situations that are not feasible with experimental techniques. For measurements, we developed an affordable testing rig to measure different types of NPs.

### 3.1. Simulation Setup

For the simulation of the sensing principle, we used a commercial finite element analysis tool, Ansys HFSS. Based on the geometry of the sensing setup and the electromagnetic properties of the materials, we extracted the impedance of the planar coil. The impedance was computed for a wide range of frequencies to find the first SRF point, which is the key value. The coil had two turns of 150μm track width and separation and a length of 12mm. The sample was a cuboid of 5mm×1mm×0.25mm. The geometry of the planar circuit and sample are schematically shown in Figure 3 (left). We used a lumped port model to excite the planar inductor. The coil’s material was copper, whose properties were included in the simulation. The environment was introduced with arbitrary magnetic permeability and electric permittivity, which could be adjusted, and the sample with its electromagnetic properties. The boundary of the simulation perfectly absorbed the incident radiation-removing reflections. All simulations were performed with 1000 frequency points in a linear range between 0 and 3GHz.

### 3.2. Experimental Setup

We performed the experimental measurements using a NanoVNA V2 calibrated at the end of the connection cables and derived the equivalent impedance of the coil from the scattering parameters. We made a linear interpolation between the first couple of positive and negative imaginary impedance values from the coil’s impedance to extract the SRF. This impedance was measured with 1000 frequency points in a linear from 100kHZ to 3GHz. The design of the sensor produced for testing matched the geometry of the simulated one, as shown in Figure 4, in which one can also see an adapter connecting it to NanoVNA V2.

### 3.3. Nanoparticles for Testing

We used two materials to test the behaviour observed in the simulations: superparamagnetic magnetite and gold colloidal suspensions. We chose Fe${}_{3}$O${}_{4}$ due to its high magnetic permeability and electric permittivity. The suspensions were chemically synthesised by co-precipitating Fe(II/III) salts. This route is one of the most straightforward synthesis methods to obtain large amounts of magnetic material. Briefly, a 100 mL solution of 27% (*w*/*v*) of Fe(III) was prepared in a beaker. Consecutively, 12.94 g of Fe(II) were dissolved in 45 mL of the total volume in a graduated cylinder. A few drops of 37% HCl were added to both solutions to ensure the perfect dissolution of the salts and avoid the possible oxidation of iron cations. Then, both solutions were mixed and stirred vigorously. For the NPs to precipitate, a basic solution was added. The drastic change in pH allows the nucleation and growth of the NPs. To have some control of these two processes, the precipitating agent must be added slowly. Therefore, a solution of 75 mL of 25% ammonia was slowly poured. Finally, the NPs obtained were magnetically decanted to eliminate the residues of the reaction three times. Then, the NPs were resuspended in distilled water.

The average NP size was 12 nm. The permeability and permittivity values of these MNPs were not measured. Still, similar ones appeared in the literature [25,26] presenting values of the relative permeability $\mu_{\mathrm{MNPs}}^{\prime }$ at RF between 1.5 and 2 and relative electric permittivity $\varepsilon_{\mathrm{MNPs}}^{{}^{\prime }}$ between 6 and 8. The gold NPs were a commercial colloid of spherical particles with a diameter of 5.6 nm. Their electric permittivity was not measured, but some studies [27] report $\varepsilon_{\mathrm{GNPs}}^{{}^{\prime }}$ values between 1.5 and 3 in similar particles. The NPs were deposited on blotting paper using a custom printer that drops a controlled flow of solution along a line pattern. Printing was performed with a flow of 3.03μL/s at a speed of 33mm/s. All the blotting paper pieces had the same dimensions 80mm×25mm, and their shape was determined to provide a uniform pressure distribution on the sensor. The shape of the lines of deposited NPs was a rectangle of 25mm×2mm. Figure 5 shows the pictures of some of the tested samples.

The blotting paper had to be flattened against the sensing surface to eliminate ripples, thus helping to keep the NP deposition at the same distance in every measurement. We achieved this by applying pressure onto the blotting paper. The clamping force, which compresses the paper, impacts the environment’s refraction index. Keeping it constant between measurements is critical to improving repeatability. For this purpose, we fabricated a custom sample holder to fix the samples firmly on top of the sensing coil. The sample holder had a plunger through which pressure was applied on both sides of a Teflon plate on top of the sensing area. The symmetric pressure resulted in a uniform tension distribution on the plate and, thus, a repeatable and predictable contact. To ensure this tension distribution, we used a 4-bar mechanism that kept the symmetric plunger parallel to the sensing circuit board. The pressure jig is shown in Figure 6.

## 4. Results and Discussion

### 4.1. Simulation Results

First, we tested the sensitivity of the simulated setup with different ideal NPs to verify the importance of the electric permittivity and its impact on the drift of the coil’s SRF. Specifically, the following parameter values were used: $\mu_{\mathrm{NPs}}=1,\;\varepsilon_{\mathrm{NPs}}=10$; $\mu_{\mathrm{NPs}}=10,$ $\varepsilon_{\mathrm{NPs}}=1$; and $\mu_{\mathrm{NPs}}=\varepsilon_{\mathrm{NPs}}=10$.

Figure 7 shows the dependence of the ln(SRF) on the ratio $m_{\mathrm{NP}}/m_{\max}$ for the NPs in all three cases.

With these simulation parameters, 0% mass yielded ln(SRF)=20.95. When only the electric permittivity (red circles in Figure 7) or the magnetic permeability (green triangles) were increased from their minimum unity value, the ln(SRF) monotonically decreased in the same way, as can be seen with the superposition of the two curves until 50% of mass ratio. This result proves that, as long as the refraction index was the same, regardless of whether it was $\mu_{\mathrm{NPs}}$ or $\varepsilon_{\mathrm{NPs}}$ that would change, the ln(SRF) evolved in the same way as the NP mass. When the refraction index increased, as in the curve with blue squares (Figure 7), the decrease in the ln(SRF) with the mass was accented. This simulation behaviour is consistent with our mathematical model as long as the variation in ln(SRF) was less than 0.04, which corresponds to an SRF change of approximately 50MHz (here, a mass ratio of 50%). This maximum variation in the SRF depends on the specific sensing coil, the environment, and NP properties. Under this limitation, the signal’s response is linear versus the mass, and the slope follows Equation (Equation 13). These results prove that inductive sensors, working in their SRF point, are sensitive to both magnetic permeability and electric permittivity and, for small variations in the SRF (such as those produced with small amounts of NPs), are well described using the mathematical model presented in this article. Consequently, the method can be used to take advantage of the inductive and capacitive effects in the coil to detect NPs with increased sensitivity.

One of this work’s most significant simplifying assumptions is the dismissal of the imaginary parts of the magnetic permeability and the electric permittivity. To check the impact of this simplification, we simulated the system in the presence of different NPs, specifically with loss tangents of $\tan\left(\delta_{e}\right)=\tan\left(\delta_{m}\right)=0$, $\tan\left(\delta_{e}\right)=\tan\left(\delta_{m}\right)=0.05$, and $\tan\left(\delta_{e}\right)=\tan\left(\delta_{m}\right)=0.1$, where $\delta_{m}=\mu^{\prime \prime }/\mu^{\prime }$ and $\delta_{e}=\varepsilon^{\prime \prime }/\varepsilon^{\prime }$.

Figure 8 shows that there is no effect of $\mu^{\prime \prime }$ and $\varepsilon^{\prime \prime }$ on the sensor response for a sample with $\mu_{\mathrm{NP}}^{\prime }=\varepsilon_{\mathrm{NP}}^{\prime }=10$. This allows us to conclude that neglecting the imaginary parts of μ and ε is an acceptable assumption.

The following simulation focuses on the environment’s effect. Manufacturing an actual sensor involves using materials whose refraction index significantly influences its behaviour. We performed simulations with the expected values of the environment permeabilities and permittivities in three cases: when the coil is in vacuum, $\mu_{\mathrm{env}}=\varepsilon_{\mathrm{env}}=1$; for the coil on top of a fibreglass material, for which we assumed $\mu_{\mathrm{env}}=1$ and $\varepsilon_{\mathrm{env}}=3$; and if the sensor is surrounded by fibreglass, we used $\mu_{\mathrm{env}}=1$ and $\varepsilon_{\mathrm{env}}=4$. In all cases, we assumed that no ferro- or ferri-magnetic material, apart from the sample, was near the coil.

For the simulation with the same NP parameters as before, $\mu_{\mathrm{NPs}}=\varepsilon_{\mathrm{NPs}}=10$, we obtained the results shown in Figure 9.

Using Equation (Equation 12), we fitted the simulated data in all three cases, thus obtaining the parameters shown in Table 1.

In this simulation, the NPs were the same, so the differences in $A_{1}$ and $A_{0}$ values came exclusively from the variation in the environment’s refraction index. Based on the $R^{2}$ values, it could be inferred that the model’s validity was compromised when the difference between the environment and the NP refraction indexes significantly increased. This problem resulted from trying to accommodate large variations in the SRF against our model’s premises. When $n_{\mathrm{env}}$ increases $A_{1}$, the line slope’s absolute value decreases, thus decreasing the sensitivity of the measurement.

### 4.2. Experimental Results

Figure 10 shows the results of measuring gold (red circles) and magnetite (green triangles) NPs with our sensor to validate the mathematical model (Equation 12). The corresponding linear regression parameters are shown in Table 2.

We can observe some differences between the experimental (Table 2) and simulated (Table 1) $A_{0}$ values. This is due to the change in the LC product (see Equation (14)) associated with the coil geometry (more precisely to the adapter, which can be seen in Figure 4). Regardless, these results prove that, despite the low concentration of NPs, the magnetite NPs exhibit five times the signal of the gold ones, with the same mass. This is evidenced by the sensitivity values $A_{1}$ and easily explained by the higher refraction index of MNPs. This difference also explains the low $R^{2}$ value of the gold. This result validates our analysis of the sensor based on changes in the SRF and confirms the possibility of easily quantifying the MNP mass in a test sample.

Considering the significant differences between this method and previous ones using both the self-inductance and self-capacitance of the inductive sensor, even when the purpose of the study is not to optimise the sensor’s characteristics but to evaluate the mathematical and measurement method, there is a remarkable improvement in the sensitivity *S*, which is multiplied by 1940, S=5821.4 Hz/μg Fe${}_{3}$O${}_{4}$ compared to S=3 Hz/μg Fe${}_{3}$O${}_{4}$ achieved in previously reported ones [9]. Even if we normalise the sensitivity by dividing it by the operating frequency in the absence of NPs, the improvement is 2.04 times. This also shows the importance of measuring at higher frequencies, which implies including the sensor’s capacitive effects and the sample’s electric permittivity.

## 5. Conclusions

Inductive sensors are affected by their environment’s electric permittivity and magnetic permeability. The importance of both factors increases as the working frequency approaches the coil’s SRF. For this reason, measuring the SRF of the sensing coil in the presence of NPs can be used to quantify them with increased sensitivity. The refraction index ratio between NPs and the whole environment determines the sensor sensitivity. In the case of MNPs, their higher magnetic permeability leads to better sensitivity, as it increases the refraction index ratio $n_{\mathrm{NPs}}/n_{\mathrm{env}}$.

In this work, we established some hypotheses under which our sensor exhibits a linear regime. This idea allowed us to calibrate the sensor using a linear fit from a sample set of known masses. This approach is limited to low-loss samples and small variations in the SRF. Using the coil’s SRF combined with our mathematical model opens the way to develop an affordable, portable, and reliable sensor to detect and quantify small numbers of MNPs. In future works, for example, it can be applied to develop an optimised magnetic lateral-flow immunoassay reader that provides qualitative and quantitative results without compromising the main advantages of rapid paper tests.

## Figures and Tables

**Figure 1 sensors-23-02372-f001:**
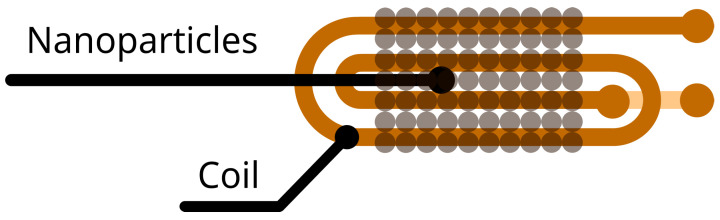
Representation of the sensing coil used in this work and sample placement.

**Figure 2 sensors-23-02372-f002:**
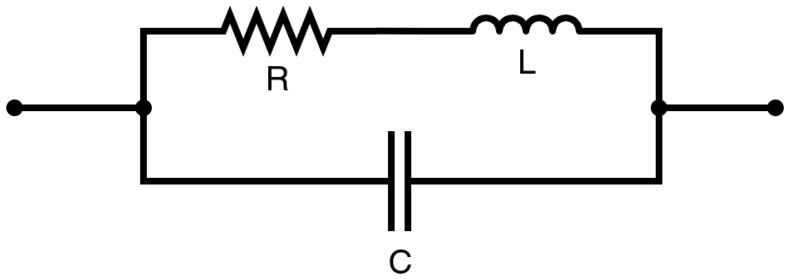
Equivalent circuit used to describe the behaviour of the inductor.

**Figure 3 sensors-23-02372-f003:**
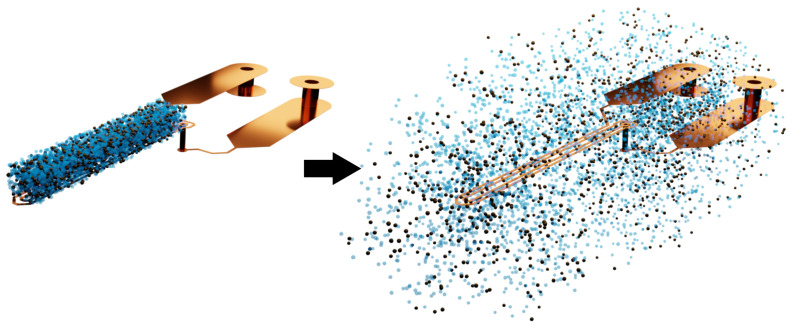
(**Left**) schematic representation of the sample (NPs and their embedding matrix) on top of the planar coil (here, the surrounding electronics and materials are not depicted) as used in experimental measurements and simulations; (**right**) scheme of the uniform particle distribution surrounding the coil as used in the mathematical model, with the same NPs and matrix amounts.

**Figure 4 sensors-23-02372-f004:**
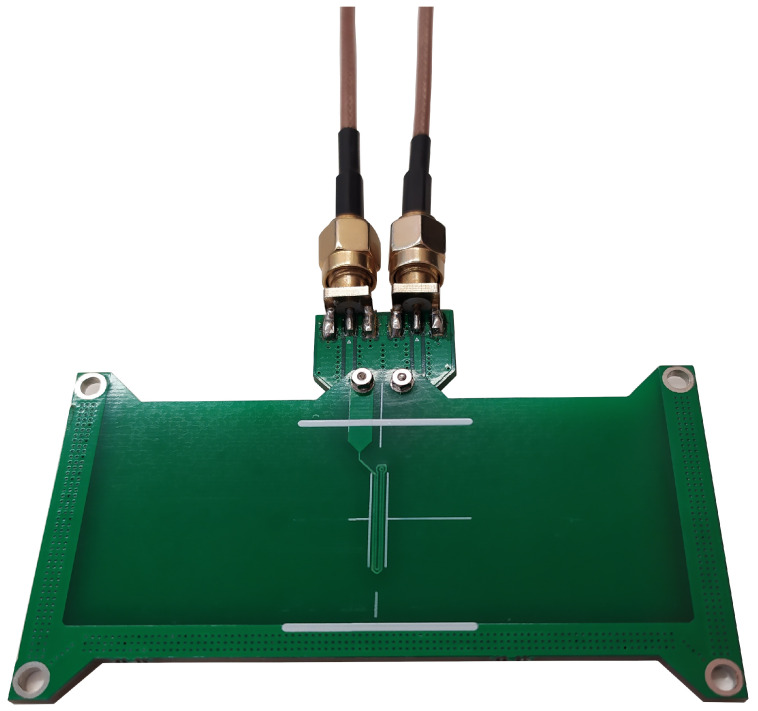
Adapter and planar coil on the printed circuit board used for the experimental measurements.

**Figure 5 sensors-23-02372-f005:**
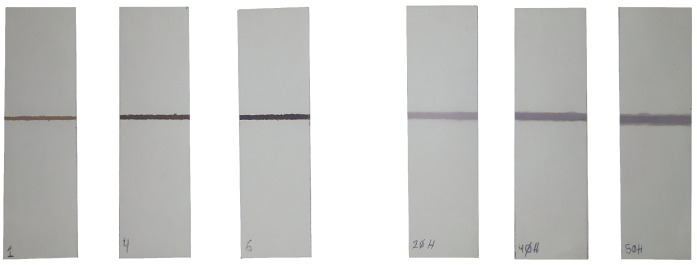
(**Left**) some magnetite NP samples; (**right**) some gold NPs samples. In all cases, NPs were deposited on blotting paper and placed on the sensing coil in the experimental setup.

**Figure 6 sensors-23-02372-f006:**
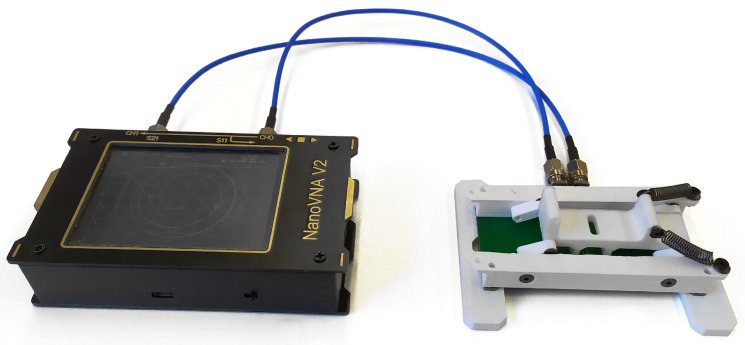
Pressure jig in combination with NanoVNA V2.

**Figure 7 sensors-23-02372-f007:**
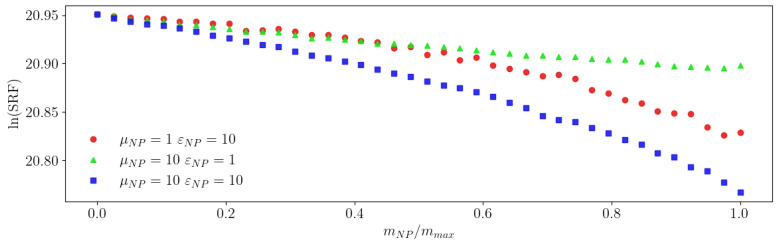
Dependence of the ln(SRF) on the ratio $m_{\mathrm{NP}}/m_{\max}$ for the simulated NPs.

**Figure 8 sensors-23-02372-f008:**
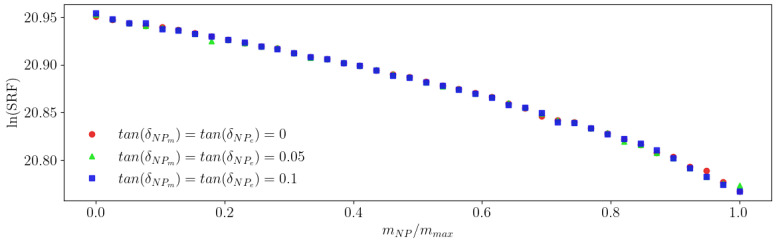
Dependence of the ln(SRF) on the ratio $m_{\mathrm{NP}}/m_{\max}$ for different lossy NPs.

**Figure 9 sensors-23-02372-f009:**
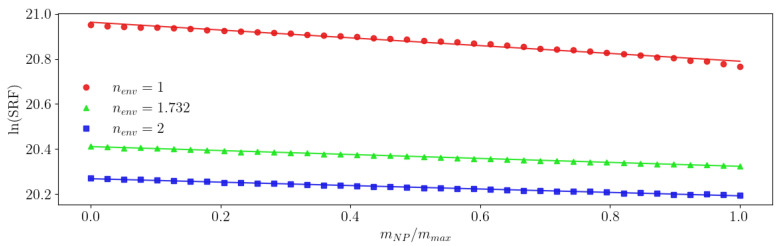
Dependence of the ln(SRF) on the ratio $m_{\mathrm{NP}}/m_{\max}$ for several environment refraction indexes. The solid lines are their fitness to our model (Equation 12).

**Figure 10 sensors-23-02372-f010:**
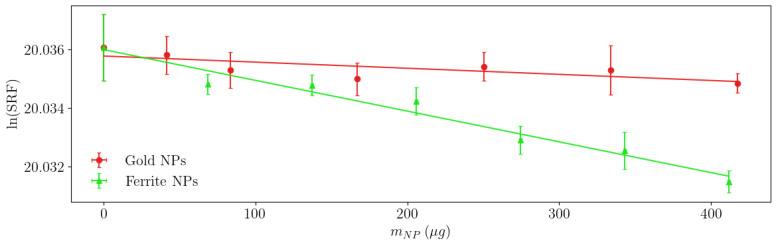
Measurements with the experimental setup (symbols) and model adjustment (lines).

**Table 1 sensors-23-02372-t001:** Fitting to Equation (Equation 12) of the simulation results for several refraction indexes in the environment.

	$A_{1}$	$A_{0}$	$R^{2}$
$n_{\mathrm{env}}=1$	$-1.7313\times 10^{-1}$	−20.9629	0.9776
$n_{\mathrm{env}}=1.732$	$-8.7557\times 10^{-2}$	−20.4109	0.9981
$n_{\mathrm{env}}=2$	$-7.6000\times 10^{-2}$	−20.2684	0.9940

**Table 2 sensors-23-02372-t002:** Fitting results of the experimental data.

	$A_{1}$	$A_{0}$	$R^{2}$
Gold NPs	$-2.0871\times 10^{-6}$	−20.0358	0.5815
Magnetite NPs	$-1.0493\times 10^{-5}$	−20.0360	0.9651

## Data Availability

Not applicable.

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
