# Peer review of "New Perspective on Planar Inductive Sensors: Radio-Frequency Refractometry for Highly Sensitive Quantification of Magnetic Nanoparticles"

_sensors, 2023, doi:10.3390/s23052372_

Round 1
Reviewer 1 Report
The manuscript entitled described “New Perspective of Planar Inductive Sensors: Radio-Frequency Refractometry for Highly Sensitive Quantification of Magnetic Nanoparticles” described new approach of diagnostics using nanoparticles.
#Lateral flow immunoassays (LFIAs) have proven to be one of the most effective methods for the quick detection of specific biomolecules. The limitations of the method in terms of sensitivity and accurate quantification have also been brought to light by this application.
# These limitations can be overcome by employing magnetic nanoparticles (MNPs) as reporter markers that are selectively attached to the biomolecule of interest. This can be done in place of the more traditional use of gold, silver, or latex particles.
#In general, detecting minor impedance changes requires expensive equipment, thus solutions with reduced costs would be perfect if they were combined with LFIAs, which are quite inexpensive.
#In this manuscript authors have identified and measured magnetic nanoparticles form that make it possible to achieve significant advancements in many fields, such as biomedicine, food safety, and environmental regulation.
# The authors have stated that If magnetic nanoparticles are used as a rapid diagnostic tests such as lateral flow immunoassays, there is ample possibilities to increase the sensitivity and quantifying capacities if the magnetic nanoparticles are paired with a sensor that matches their primary benefits, which are low cost, portability, and user-friendliness.
# Nevertheless, they have a restricted sensitivity that is insufficient for the minuscule number of particles that can be found in a lateral flow strip. The frequency at which coils self-resonate is extremely sensitive to changes in magnetic permeability as well as electric permittivity, or more technically, the particle refraction index.
#The considerable complexity of the modelling required to attain the requisite quantification is one of its drawbacks. The authors used an equivalent impedance model and a focused hypothesis that was particular to nano-disperse materials in order to produce a simpler model that is easy to use and maintains the method's advantages.
#This model was built as part of this body of work and validated by analysing the results of 3D electromagnetic simulations as well as experimental data.
# This model has been validated by comparison to the results of simulations and experiments in order to determine the scope of its applicability and identify any gaps in its coverage.
#Finally, authors have demonstrated that the developed model is feasible to calibrate a sensor for a particular class of nano-particles in order to do accurate mass measurement at a relatively low cost.
In light of what has been stated above, the manuscript appears to be fascinating and could be worthwhile if it were to be published. The research was conducted correctly and was written in clear English. Nevertheless, there is room for development in the English language so that even a layperson can comprehend the model.
Author Response
We are grateful to receive feedback on our manuscript submitted to Sensors. We have revised it accordingly.
We greatly appreciate the reviewers’ comments that have helped us to improve the manuscript. Below, we list our answers and corrections.
To Reviewer #1
Recommendation 1: The research was conducted correctly and was written in clear English. Nevertheless, there is room for development in the English language so that even a layperson can comprehend the model.
Answer 1: Done. The language has been revised, and additional information has been added to the manuscript
Reviewer 2 Report
Can the author compare the sensitivity of the present device with the already available one, in a table, this could be of great significance!
Author Response
We are grateful to receive feedback on our manuscript submitted to Sensors. We have revised it accordingly.
We greatly appreciate the reviewers’ comments that have helped us to improve the manuscript. Below, we list our answers and corrections.
To Reviewer #2
Recommendation 1: Can the author compare the sensitivity of the present device with the already available one, in a table, this could be of great significance!
Answer 1: Done. The comparison of our sensor with others has been presented in the text (lines 276-284):
”Considering the important differences of this method compared to previous ones, using the self-inductance and self-capacitance of the inductive sensor (even when this work’s purpose is not to optimize the sensor’s characteristics but evaluate the mathematical and measurement method) leads to a sensitivity S improvement of 1940 times, S = 5821.4 Hz /μg Fe3O4 compared to S = 3 Hz /μg Fe3O4 achieved in previously reported ones [9 ]. Even if we normalize the sensitivity by dividing it by the frequency in the absence of NPs, the improvement is 2.04 times. This also shows the importance of measuring at higher frequencies, which implies including the sensor’s capacitive effects and the sample’s electric permittivity.”
Reviewer 3 Report
Report on sensors
Manuscript title: " New Perspective of Planar Inductive Sensors: Radio-Frequency
Refractometry for Highly Sensitive Quantifiation of Magnetic Nanoparticles".
There are some minor comments considered as follows:
1- The paper should be reviewed linguistically and grammatically checked for the paper.
2-The main idea of ​​the study must be clarified.
3-The introduction should be reformulated in terms of the sequence and arrangement of literary ideas.
4- Future research direction most be shown in conclusion.
5- The discussion is not sufficient to describe the results of the phenomenon.
Author Response
We are grateful to receive feedback on our manuscript submitted to Sensors. We have revised it accordingly.
We greatly appreciate the reviewers’ comments that have helped us to improve the manuscript. Below, we list our answers and corrections.
To Reviewer #3
Recommendation 1: The paper should be reviewed linguistically and grammatically checked for the paper.
Answer 1: Done . A native English speaker has revised and corrected the English language.
Recommendation 2: The main idea of the study must be clarified.
Answer 2: Done . The introduction has been rewritten to clarify the main idea, for example in lines (20-57).
”New sensors are needed to detect and quantify nanoparticles (NPs) for increasing new applications. One example is the detection of biomolecules (toxins, disease biomarkers or others), which are selectively attached to the particles by an immunological reaction [1]. There are several options for it, such as immobilizing the particle-labelled molecule onto the sensor surface, using microfluidics to take the sample to the sensor or to pass it over its sensitive part, or the paper-based microfluidics [2,3]. The latter is well known for being the basis of the COVID-19 rapid diagnostic tests [4]. Such tests have two main limitations that hinder their wider application; One is the lack of sensitivity (many fake negative results), and the other one is their unreliable quantification. To improve upon it, some authors rely on magnetic nanoparticles (MNPs) as labels [5–8], which can be detected by devices sensitive to the fringes of their magnetic field, as in [9]. New needs in Life Sciences, such as healthcare, food safety, and environmental control, require further improvement of the sensitivity and the quantifying capabilities of the MNP-detecting sensors due to the extremely small number of particles immobilized at the test line and their tiny size. Previous works [10–12] stated the ability to detect and quantify small numbers of MNPs by monitoring the impedance changes of a planar coil at a fixed frequency in the order of tens of MHz. The sensor is sensitive to the magnetic permeability variations around the planar coil [13] as a consequence of Faraday’s electromagnetic induction law and can be observed as changes in its impedance. The larger the frequency, the larger the impedance change, as far as we are below the coil’s resonance frequency. In general, reliably obtaining small impedance changes in the MHz range requires expensive equipment (such as impedance or vector network analyzers,) so lower-cost and portable solutions would be more accessible and better combined with the inexpensive LFIAs. Some reported miniaturized devices focus on cost reduction using oscillator-based inductive detectors, their reported sensitivity being 3 Hz/μg Fe3 O4 [9,14]. Such sensitivity is not enough for the requirements of many new potential applications, like toxin detection in foods or early diagnosis. Based on the central idea of [14], in this work, we have improved the MNPs detection sensitivity while keeping costs down using the sensor coils’ self-resonant frequency (SRF). The main roadblock of SRF-based detectors is the complexity of the mathematical relation between the SRF and the MNPs’ mass. We have solved this bottleneck as described in section 2. A usable, easy-to-implement and precise mathematical model to correlate the coils’ SRF and the nanoparticles’ mass (volume or number, depending on the variable used for the calibration) is indispensable to make viable this measurement technique. We provide such a targeted model and test it against simulations and experimental results to verify its validity and explore its limitations. Even when the central intention of this article is to develop and test the mathematical model, we have proved a significant sensitivity increase and reduction of the signal-to-noise ratio.”
Recommendation 3: The introduction should be reformulated in terms of the sequence and arrangement of literary ideas.
Answer 3: Done. The introduction has been reformulated to reflect better the arrangements of literary ideas, lines (20-57):
“New sensors are needed to detect and quantify nanoparticles (NPs) for increasing new applications. One example is the detection of biomolecules (toxins, disease biomarkers or others), which are selectively attached to the particles by an immunological reaction [1]. There are several options for it, such as immobilizing the particle-labelled molecule onto the sensor surface, using microfluidics to take the sample to the sensor or to pass it over its sensitive part, or the paper-based microfluidics [2,3]. The latter is well known for being the basis of the COVID-19 rapid diagnostic tests [4]. Such tests have two main limitations that hinder their wider application; One is the lack of sensitivity (many fake negative results), and the other one is their unreliable quantification. To improve upon it, some authors rely on magnetic nanoparticles (MNPs) as labels [5–8], which can be detected by devices sensitive to the fringes of their magnetic field, as in [9]. New needs in Life Sciences, such as healthcare, food safety, and environmental control, require further improvement of the sensitivity and the quantifying capabilities of the MNP-detecting sensors due to the extremely small number of particles immobilized at the test line and their tiny size. Previous works [10–12] stated the ability to detect and quantify small numbers of MNPs by monitoring the impedance changes of a planar coil at a fixed frequency in the order of tens of MHz. The sensor is sensitive to the magnetic permeability variations around the planar coil [13] as a consequence of Faraday’s electromagnetic induction law and can be observed as changes in its impedance. The larger the frequency, the larger the impedance change, as far as we are below the coil’s resonance frequency. In general, reliably obtaining small impedance changes in the MHz range requires expensive equipment (such as impedance or vector network analyzers,) so lower-cost and portable solutions would be more accessible and better combined with the inexpensive LFIAs. Some reported miniaturized devices focus on cost reduction using oscillator-based inductive detectors, their reported sensitivity being 3 Hz/μg Fe3 O4 [9,14]. Such sensitivity is not enough for the requirements of many new potential applications, like toxin detection in foods or early diagnosis. Based on the central idea of [14], in this work, we have improved the MNPs detection sensitivity while keeping costs down using the sensor coils’ self-resonant frequency (SRF). The main roadblock of SRF-based detectors is the complexity of the mathematical relation between the SRF and the MNPs’ mass. We have solved this bottleneck as described in section 2. A usable, easy-to-implement and precise mathematical model to correlate the coils’ SRF and the nanoparticles’ mass (volume or number, depending on the variable used for the calibration) is indispensable to make viable this measurement technique. We provide such a targeted model and test it against simulations and experimental results to verify its validity and explore its limitations. Even when the central intention of this article is to develop and test the mathematical model, we have proved a significant sensitivity increase and reduction of the signal-to-noise ratio.”
Recommendation 4: Future research direction must be shown in the conclusion.
Answer 4: Done. Future research ideas have been added to the conclusion section in lines 298-300.
“In future works, for example, it can be applied to develop an optimized magnetic lateral flow immunoassay reader offering qualitative and quantitative results without compromising the main advantages of the rapid paper test.”
Recommendation 5: The discussion is not sufficient to describe the results of the phenomenon.
Answer 5: Done. We have extended the discussion considerably to clarify the phenomenon better. The most significant changes are in lines 219-236, 256-262, and 267-284.
“With these simulation parameters, 0% mass yields ln(SRF) = 20.95. When only the electric permittivity (red circles in figure 7) or the magnetic permeability (green triangles) are increased from their minimum unity value, the ln( SRF) decreases monotonically in the same way, as can be seen by the superposition of the two curves until 50% of mass ratio. This result proves that, as far as the refraction index is the same, no matter if it is μNPs or εNPs which changes, the ln(SRF) evolves in the same way with the NP mass. When the refraction index increases, as in the curve with blue squares (figure 7), the decrease of the ln(SRF) with the mass is accented. The simulation behaviour is compliant with our mathematical model as far as the variation of ln(SRF) is less than 0.04, which corresponds to an SRF change of approximately 50 MHz (here a mass ratio of 50%). This maximum variation of the SRF depends on the specific sensing coil, the environment, and the NPs properties. Under this limitation, the signal’s response is linear versus the mass, and the slope follows equation (13). These results prove that inductive sensors, working in their SRF point, are sensitive to both magnetic permeability and electric permittivity and, for small variations of the SRF (as those produced by small amounts of NPs), and well described by the mathematical model presented in this article. Consequently, the method can be used to take advantage of the inductive and capacitive effects in the coil to detect the NPs with increased sensitivity.”
“In this simulation, the NPs are the same, so the differences in A1 and A0 values come exclusively from the environment’s refraction index variation. Based on the R2 values, one can assess that the model’s validity is compromised when the difference between the environment and the NPs refraction indexes increases too much. This problem comes from trying to accommodate large variations of the SRF against our model’s premises. When nenv increases A1 , the line slope absolute value decreases, thus decreasing the sensitivity of the measurement.”
“We can observe some differences between the experimental (table 2) and simulated (table 1) A0 values. This is due to the change of the LC product (see equation 14) associated with the coil geometry (more precisely to the adapter, which can be seen in figure 4). Regardless, these results prove that, despite the NP low concentration, the magnetite NPs exhibit five times the signal of the gold ones, with the same mass. This is evidenced by the sensitivity values A 1 and easily explained by the MNPs higher refraction index. This difference also explains the low R2 value of the gold. This result validates our analysis of the sensor based on changes in the SRF and confirms the possibility of easily quantifying the MNP mass in a test sample. Considering the significant differences of this method compared to previous ones, using both the self-inductance and self-capacitance of the inductive sensor, even when this work’s purpose is not to optimize the sensor’s characteristics but evaluate the mathematical and measurement method, there is a remarkable improvement in the sensitivity S, which is multiplied by 1940, S = 5821.4 Hz/μg Fe3 O4 compared to S = 3 Hz/μg Fe3 O4 achieved in previously reported ones [9]. Even if we normalize the sensitivity by dividing it by the operating frequency in the absence of NPs, the improvement is 2.04 times. This also shows the importance of measuring at higher frequencies, which implies including the sensor’s capacitive effects and the sample’s electric permittivity.”